# Placental Mesenchymal Dysplasia and Beckwith–Wiedemann Syndrome

**DOI:** 10.3390/cancers14225563

**Published:** 2022-11-12

**Authors:** Hidenobu Soejima, Satoshi Hara, Takashi Ohba, Ken Higashimoto

**Affiliations:** 1Division of Molecular Genetics and Epigenetics, Department of Biomolecular Sciences, Faculty of Medicine, Saga University, Saga 849-8501, Japan; 2Department of Obstetrics and Gynecology, Faculty of Life Sciences, Kumamoto University, Kumamoto 860-8556, Japan

**Keywords:** placental mesenchymal dysplasia, Beckwith–Wiedemann syndrome, genomic imprinting, DNA methylation, differentially methylated regions, androgenetic/biparental mosaicism (paternal uniparental diploidy/biparental diploidy mosaicism)

## Abstract

**Simple Summary:**

Placental mesenchymal dysplasia (PMD) is a morphological abnormality resembling partial hydatidiform moles without abnormal trophoblastic proliferation. In PMD, approximately 20% of fetuses have Beckwith–Wiedemann syndrome (BWS), and approximately 20% of BWS fetuses are associated with PMD. In addition, PMD is a cardinal feature of BWS, and paternal uniparental diploidy/biparental diploidy mosaicism (also called androgenetic/biparental mosaicism) has been found in both BWS and PMD. This suggests that there is a molecular link between BWS and PMD. In this review, we focus on the etiologies of BWS and PMD and describe the molecular link between them. Both conditions are imprinting disorders that, depending on the case, may share or differ in molecular characteristics. These observations raise questions concerning the timing of the occurrence of the molecularly abnormal cells during the postfertilization period and the effects of these abnormalities on cell fates after implantation.

**Abstract:**

Placental mesenchymal dysplasia (PMD) is characterized by placentomegaly, aneurysmally dilated chorionic plate vessels, thrombosis of the dilated vessels, and large grapelike vesicles, and is often mistaken for partial or complete hydatidiform mole with a coexisting normal fetus. Androgenetic/biparental mosaicism (ABM) has been found in many PMD cases. Beckwith–Wiedemann syndrome (BWS) is an imprinting disorder with complex and diverse phenotypes and an increased risk of developing embryonal tumors. There are five major causative alterations: loss of methylation of imprinting control region 2 (*KCNQ1OT1*:TSS-DMR) (ICR2-LOM), gain of methylation at ICR1 (*H19*/*IGF2*:IG-DMR) (ICR1-GOM), paternal uniparental disomy of 11 (pUPD11), loss-of-function variants of the *CDKN1C* gene, and paternal duplication of 11p15. Additional minor alterations include genetic variants within ICR1, paternal uniparental diploidy/biparental diploidy mosaicism (PUDM, also called ABM), and genetic variants of *KCNQ1*. ABM (PUDM) is found in both conditions, and approximately 20% of fetuses from PMD cases are BWS and vice versa, suggesting a molecular link. PMD and BWS share some molecular characteristics in some cases, but not in others. These findings raise questions concerning the timing of the occurrence of the molecularly abnormal cells during the postfertilization period and the effects of these abnormalities on cell fates after implantation.

## 1. Introduction

Placental mesenchymal dysplasia (PMD) was first reported by Moscoso [1] as diffuse mesenchymal hyperplasia of the placental stem villi, leading to increased placental volume (placentomegaly) and elevated levels of alpha fetoprotein (AFP). It is now known to be characterized by varying levels of placentomegaly, aneurysmally dilated chorionic plate vessels, thrombosis of the dilated vessels, and large grapelike vesicles within the placenta [2]. Since these features may mimic a molar pregnancy on ultrasound, it is often mistaken for partial hydatidiform mole (PHM) or complete hydatidiform mole (CHM) with a coexisting normal fetus [3]. Unlike molar pregnancies, PMD usually features a normal fetus and the pregnancy often extends into the third trimester [4]. However, PMD pregnancies are high-risk because they carry fixed probabilities of several complications. These include fetal growth restriction (FGR); preterm delivery; fetal demise; preeclampsia; eclampsia; hemolysis, elevated liver enzymes, and low platelets (HELLP) syndrome; and hypertensive disorders of pregnancy (HDP) [2,3,5,6]. In addition, approximately 20% of fetuses in PMD pregnancies have Beckwith–Wiedemann syndrome (BWS) [4]. Since androgenetic/biparental mosaicism (ABM) and androgenetic/biparental chimera (ABC) have been found in PMD specimens, the presence of androgenetic cells resulting in abnormal genomic imprinting has been suggested as a cause of PMD [7,8,9,10].

BWS (OMIM #130650) was originally reported by Beckwith and Wiedemann [11,12] as a syndrome involving exomphalos, macroglossia, and gigantism. It is now known to be an imprinting disorder with complex and diverse phenotypes. Macroglossia, exomphalos, and lateralized overgrowth are the cardinal features, and there is also an increased risk of developing embryonal tumors such as Wilms tumor, hepatoblastoma, neuroblastoma, rhabdomyosarcoma, adrenocortical carcinoma, or phaeochromocytoma [13,14]. Because of the range of phenotypes, it is recommended that the Beckwith–Wiedemann spectrum (BWSp) is used and that diagnoses are made based on a scoring system. This system assigns two points to each cardinal feature and one point to each suggestive feature. Patients with a total score of ≥4 are diagnosed as classical BWS, irrespective of their molecular test results, as are patients with a score of ≥2 with positive genetic test results [13]. Placentomegaly and PMD are included as the suggestive or cardinal feature, respectively, in the BWSp scoring system [13]. There are five major causative alterations: loss of methylation of imprinting control region 2 (*KCNQ1OT1*:TSS-differentially methylated region (DMR)) (ICR2-LOM), gain of methylation at ICR1 (*H19*/*IGF2*:IG-DMR) (ICR1-GOM), paternal uniparental disomy of 11 (pUPD11), loss-of-function variants of the *CDKN1C* gene, and paternal duplication of 11p15 [13]. The genetic testing aims to detect these major alterations. Several minor alterations have also been discovered thus far: genetic variants within ICR1, paternal uniparental diploidy/biparental diploidy mosaicism (PUDM, also called ABM), and genetic variants of the *KCNQ1* gene [15,16,17,18,19].

ABM (also called PUDM) has been found in both BWS and PMD [8,17], and approximately 20% of fetuses from PMD cases are BWS [4,5,6,20]. This suggests that there is a molecular link between BWS and PMD. In this review, we summarize the incidence, pathology, and clinical findings of PMD and its major complications, such as BWS and hepatic mesenchymal hamartoma. In addition, we focus on the etiologies of BWS and PMD and describe the molecular link between them.

## 2. Incidence and Pathology of PMD

### 2.1. Incidence

In 2001, Paradinas et al. reported the incidence of PMD as 0.2% (15 out of 7560 placentas examined) [21]. The following year, Arizawa et al. reported an incidence of only 0.02% (7/30,758) [22], and in 2012, Zeng et al. reported an incidence of 0.002% (2/95,265) [23]. Although the intermediate figure (0.02%) is widely used in the literature, calculation of the true incidence may be difficult, since only a small fraction of placentas are subjected to pathologic examination [3]. In any case, PMD is certainly a rare placental condition, and so far only just over 100 cases have been reported [3,24].

More than 80% of the fetuses from PMD pregnancies identified thus far have been female with the normal karyotype (46,XX) [3,4,6]. However, a karyotype of 46,XX/46,XY has been observed in some fetuses. Cohen et al. reported a normal karyotype in 32 of 36 cases (89%) and an abnormal karyotype in the remaining four (11%) [20]. The abnormal karyotypes in that study included trisomy 13, 47,XXY (Klinefelter syndrome), and 69,XXX (triploidy), and these karyotypes were confirmed in the placental specimens. In addition, 13q12.11 deletion in a neonate and another case of trisomy 13 in a fetus have been reported, but these were not confirmed in PMD specimens [25,26].

### 2.2. Pathology

Macroscopically, PMD is usually characterized by placentomegaly, meaning that the weight of the placenta is greater than the 95th percentile of placenta weights [27], and is often associated with dilatation and congestion of the vessels in the chorionic plate. In some cases, grape-like cysts formed by dilatation of the hydropic stem villi are present (Figure 1A) [5]. Most placentas with PMD have two distinct areas: one with a macroscopically normal appearance and one exhibiting macroscopic PMD characteristics (a cystic lesion) (Figure 1A) [3,28]. In a small proportion of cases the macroscopic PMD lesion occupies the entire placenta.

The microscopic features of PMD include enlarged stem-cell villi with varying degrees of edema and containing abnormal thick-walled vessels, which in some cases are thrombosed (Figure 1B). Trophoblastic proliferation and stromal inclusions, both characteristic of molar pregnancies, are absent (Figure 1B) [5]. The product of the maternally expressed imprinted gene *CDKN1C*, p57^KIP2^, is strongly expressed in the cytotrophoblasts and villous mesenchyme in normal placentas, but in CHM it is absent or markedly reduced, regardless of whether it is androgenetic or biparental CHM (BiCHM) [29,30,31]. In PMD, p57^KIP2^ expression is absent in the stromal cells (Figure 1C), which contain the androgenetic genome, but normal in the villous cytotrophoblast cells, which contain the biparental genome, leading to normal proliferation [8,32,33].

## 3. Clinical Findings in PMD

### 3.1. Ultrasound Findings

In ultrasound imaging, PMD is usually detected as a cystic placenta that has hypoechoic areas (80%), is enlarged or thickened (50%), and has dilated chorionic villi (16%) (Figure 1D) [5]. The majority of 49 PMD cases showed a thickened chorionic plate with a multicystic lesion that resembled PHM or CHM with co-twin during the first half of the pregnancy [6]. The course of the cystic lesions was varied: in some cases they gradually became more apparent and in others they disappeared [6].

### 3.2. Maternal Complications and Pregnancy and Neonatal Outcome

Maternal serum α-fetoprotein (MSAFP) was elevated in 70–80% of patients with PMD, while maternal serum human chorionic gonadotropin (MShCG) levels for the gestational age were elevated in 18–38% of patients with PMD [5,6]. For prenatal diagnosis, in addition to ultrasound findings, elevated MSAFP levels with normal MShCG levels in the second and third trimesters may be indicative of PMD [6].

Nine percent of PMD cases in one study showed preeclampsia, eclampsia, HELLP, or gestational hypertension as maternal complications [5]. In another study, hypertensive disorders of pregnancy (HDP) were found in 12.8% of PMD cases [6].

Fetal growth restriction (FGR), preterm delivery, and fetal demise occurred in respectively 33–72%, 53–64%, and 13–18% of cases covered in recent PMD case reviews [2,5,6]. Neonatal outcomes include BWS, hepatic tumors, and hematologic disorders (anemia and thrombocytopenia) [3,5,6]. In the following we discuss BWS and hepatic tumors.

## 4. Complications of PMD

### 4.1. BWS

BWS is frequently associated with PMD. Infants delivered from mothers with PMD were diagnosed as BWS in 12 of 66 (18.2%), 15 of 71 (21.1%), 12 of 64 (18.8%), and 8 of 47 PMD cases (17.0%), respectively, in four previous reports [4,5,6,20]. Conversely, PMD was a complication in the pregnancies of 13 of 60 neonates with BWS (21.7%) [34]. In our own data, we recorded seven PMD pregnancies among 66 neonates with BWS (10.6%). Therefore, the frequencies of BWS in PMD cases and of PMD in BWS cases are roughly similar.

Placentomegaly was observed in 70.9% and 12.0% of BWS patients in two reports, which suggests using this as a suggestive feature in the BWSp scoring system [34,35]. In the placental pathological findings in one of these studies, there was no significant difference between the causative molecular subgroups for BWS [34]. Armes et al. performed pathological examinations on eight placentas: three enlarged ones from BWS patients with ICR2-LOM, one PMD one from a BWS patient with pUPD11, and four PMD placentas without BWS features [32]. They did not find any pathological features of PMD in the three placentas from the BWS patients with ICR2-LOM, but they did find a striking excess of extravillous trophoblast. Since four of the PMD placentas exhibited ABM, and a fifth had pUPD11, they concluded that *IGF2* was a strong candidate gene for PMD.

### 4.2. Hepatic Mesenchymal Hamartoma

Hepatic mesenchymal hamartoma (HMH) is defined as the excessive, focal growth of an admixture of hepatic vascular and epithelial components, which becomes multicystic as it enlarges [36]. It is the second-most-common benign liver tumor in children [37] and one of the most commonly occurring tumors in fetuses from PMD pregnancies. An HMH case that was probably complicated with PMD, since it showed prominent dilatation of the placental vessels with multiple thromboses, was reported in 1983 [38]. Thus far, 19 HMH cases with PMD have been reported [23,36,38,39,40,41,42,43,44,45,46,47,48,49,50,51,52,53,54,55]. Of these, three cases showed ABM in both the HMH and the PMD [45,51,52], one showed ABM in the HMH but had no genotype information for the PMD [53], and one showed ABM in the PMD but had no genotype information for the HMH [55]. Based on these findings and the recurrent identification of ABM in both sporadic PMD and sporadic HMH, ABM resulting in the imbalanced expression of imprinted genes has been considered as a causative mechanism [45,52]. For example, expression of *C19MC*, which is a placenta-specific imprinted locus that encodes a cluster of 46 microRNAs (miRNAs), increased in 10 of 10 cases of sporadic HMH and three cases of HMH associated with ABM, despite the absence of expression in the normal liver [52]. In another study, cases with ABM-associated HMH and some cases of sporadic HMH harbored a hypomethylated allele of the *C19MC* differentially methylated region 1 (*C19MC*-DMR1, also called *DPRX*/*MIR512*:IG-DMR) located at its promoter region [53,56]. In addition, chromosomal rearrangement or translocation involving 19q13.4, the locus of *C19MC*, may disrupt the *C19MC* promoter and drive the expression of *C19MC* miRNAs [53]. These observations suggest the involvement of increased expression of the *C19MC* miRNA cluster as a result of hypomethylation of the *C19MC*-DMR1 (*DPRX*/*MIR512*:IG-DMR) or rearrangement of the *C19MC* promoter in the pathogenesis of HMH. However, since other cases with HMH did not express *C19MC* miRNAs, there may also be causative molecular mechanisms that are independent of *C19MC* [53].

## 5. Etiology

### 5.1. Etiology of BWS

The major etiologies of BWS are known to be ICR2-LOM (50%), pUPD11 (20%), ICR1-GOM (5%), loss-of-function variants of the *CDKN1C* gene (5%), and paternal duplication of 11p15.5 (2–4%) [13]. Other minor etiologies are single-nucleotide variants (SNVs) of or microdeletion within ICR1 [15,16,57,58], PUDM (also called ABM) [17,59,60], and genetic variants of the *KCNQ1* gene [19]. However, no molecular alteration has been found in approximately 15% of patients.

ICR2-LOM occurs on the maternal allele and induces the repression of *CDKN1C*, which encodes cyclin dependent kinase inhibitor p57^KIP2^, from the maternal allele [14,61]. The *KCNQ1* variants, including SNVs and copy number variations (CNVs), on the maternal allele cause loss of transcription of *KCNQ1* and lead to ICR2-LOM, resulting in repression of *CDKN1C* [18,19]. ICR1-GOM occurs on the maternal allele and induces the expression of *IGF2*, which encodes insulin-like growth factor 2, and the repression of *H19*, a noncoding RNA, from the maternal allele [14,61]. SNVs or microdeletion of the Oct-binding site within ICR1 are found in approximately 20% of ICR1-GOM cases and cause ICR1-GOM, resulting in biallelic expression of *IGF2* and biallelic repression of *H19* [15,16,57,58]. The extent of pUPD11 includes both ICR1 and ICR2, resulting in both ICR1-GOM and ICR2-LOM in the pUPD11 cells [60]. PUDM (ABM) is found in approximately 8% of pUPD11 cases. PUDM cases are highly susceptible to both benign and malignant tumors and should be considered at risk of autosomal recessive diseases in addition to BWS features [17,60,62]. The chromosomal region of paternal duplication of 11p15.5 is usually restricted to ICR1, resulting in overexpression of *IGF2* because there are two copies of the active gene [63]. Since epigenetic alterations such as ICR2-LOM and ICR1-GOM without genetic alterations and pUPD11 (including PUDM) are generally postzygotic events, BWS patients with these alterations are mosaics of altered and normal cells in varying ratios.

Approximately 30% of BWS patients with ICR2-LOM show multilocus imprinting disturbances (MLIDs), that is, a disturbed methylation status (often hypomethylation) at multiple imprinted differentially methylated regions (DMRs) [64]. Pathogenic variants of genes encoding components of the subcortical maternal complex (SCMC), such as *NLRP2*, *NLRP5*, *NLRP7*, *PADI6*, and *ZAR1*, have been found in patients with MLIDs and in their mothers [65,66]. The SCMC is a large multimeric protein complex in the mature mammalian oocyte that is localized at its periphery, and its components are exclusively expressed by the maternal genome [66]. The functions of the SCMC are epigenetic reprogramming and the maintenance of ploidy in the early embryo [66,67]. It is hypothesized that both homozygous and heterozygous variants of these genes cause aberrant imprinting in the offspring and lead to recurrent reproductive failure [66].

### 5.2. Etiology of PMD

#### 5.2.1. Etiology of PMD with ABM or ABC

ABM and ABC have been found in PMD specimens [7,8,9,10]. ABM can be caused by failed replication of the maternal genome after fertilization or as a result of dispermy (fertilization of one oocyte by two haploid sperm cells) (Figure 2A) [10]. In the first case, ABM occurs when the zygote divides after the paternal genome has been replicated but the maternal genome has not. The daughter cell that carries only the paternal genome gives rise to androgenetic cells by way of endoreduplication of the paternal genome. In the second case, ABM occurs when a triploid zygote formed by two sperms and one oocyte divides without replication, giving rise to a (diploid) biparental cell and a haploid cell carrying only a paternal genome. Similarly to the first case, the paternal genome is then endoreduplicated in the haploid cell, giving rise to androgenetic cells. Conversely, ABC occurs when a normal zygote fuses with one created by the fertilization of an anuclear oocyte by a sperm cell. In the latter zygote, the paternal genome is endoreduplicated, giving rise to androgenetic cells (Figure 2B). In PMD, androgenetic cells are distributed throughout the chorionic membrane, chorionic mesenchyme, stroma, and enlarged chorionic vessels, but not the trophoblast, which contains only biparental cells [8,32]. PMD trophoblasts therefore do not exhibit any abnormal proliferation, and they test positive for p57^KIP2^ (encoded by *CDKN1C*) [8,32]. The presence of ABM/ABC suggests the involvement of abnormal genomic imprinting in the etiology of PMD. This is supported by the following facts: the presence of BWS in approximately 20% of PMD cases mentioned above [4,5,6,20]; the mosaicism of the maternal deletion of 11p15.5 found in placentas with PMD [68]; the partial trisomy (two paternal copies and one maternal copy) of 11p15.5 found in an enlarged placenta with edematous villi [69]; the limitation of pUPD11 to 11p in a biparental-PMD case [32]; and the placentomegaly and dysplasia that are observed in mice with a null mutation of *Cdkn1c* and loss of *Igf2* imprinting [70]. In addition, we recently performed a DNA methylation analysis of 15 placenta-specific DMRs, which have been verified as gametic maternally methylated DMRs in several previous studies [71,72,73], and of 36 ubiquitous DMRs in PMD specimens with ABM. We found that those specimens showed the paternal epigenotype at most DMRs, including both placenta-specific and ubiquitous DMRs, especially ICR1-GOM and ICR2-LOM (Figure 3) [28]. Taken together, these facts strongly suggest that abnormal imprinting is involved in the pathogenesis of PMD with ABM, and that *IGF2* and *CDKN1C* at 11p15.5 are important molecular alterations in this pathogenesis.

#### 5.2.2. Etiology of Biparental PMD

Several PMD cases composed entirely of biparental cells have also been reported [32,74,75,76,77,78]. Recently, we genotyped 25 macroscopic PMD specimens and found that 9 (36%) showed a normal biparental genotype [28]. The abovementioned DNA methylation analysis of multiple imprinted DMRs revealed aberrant hypomethylation at seven placenta-specific DMRs (Figure 3). In this analysis, hypomethylated DMRs were defined as those that were aberrantly hypomethylated but not aberrantly hypermethylated in more than half of the specimens. Furthermore, five imprinted genes (*MCCC1*, *CRYBG1*, *AGBL3*, *GLIS3*, and *DNMT1*) associated with these DMRs were expressed biallelically [28]. Aberrant hypomethylation was also observed at ubiquitous DMRs, including *GRB10*:alt-TSS-DMR, but not at ICR1 or ICR2. *NAP1L5*:TSS-DMR and *WRB*:alt-TSS-DMR were also frequently hypomethylated in biparental-PMD specimens, but the genes associated with these DMRs may not be involved in the pathogenesis of biparental PMD because normal-sized placentas were found in mice with two paternal copies of *NAP1L5* and there was no correlation between *WRB* expression and DMR methylation in human placentas [79,80]. All of the aberrantly hypomethylated DMRs were maternally methylated under normal conditions. These results imply that the hypomethylation of DMRs, including placenta-specific DMRs and ubiquitous *GRB10*:alt-TSS-DMR, but not ICR1 or ICR2, plays a role in the pathogenesis of biparental PMD.

**Figure 3 cancers-14-05563-f003:**
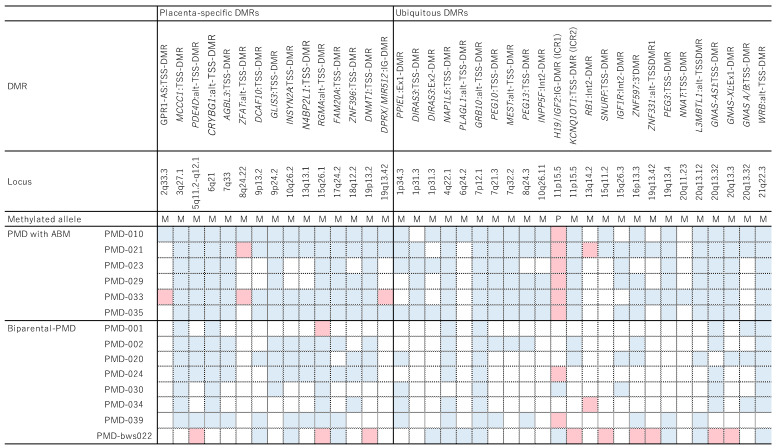
We performed a DNA methylation analysis of gametic DMRs, including 15 placenta-specific and 36 ubiquitous DMRs, in six specimens of PMD with ABM and eight biparental-PMD specimens [28]. Representative results from all 15 placenta-specific DMRs and 24 of the ubiquitous DMRs are shown. Seven placenta-specific DMRs (*MCCC1*:TSS-DMR, *CRYBG1*:alt-TSS-DMR, *AGBL3*:TSS-DMR, *GLIS3*:TSS-DMR, *INSYN2A*:TSS-DMR, *N4BP2L1*:TSS-DMR, and *FAM20A*:TSS-DMR) were aberrantly hypomethylated, which was defined as aberrantly hypomethylated but not aberrantly hypermethylated in more than half of the specimens (i.e., four or more of the biparental-PMD specimens). Aberrant hypomethylation was also observed at ubiquitous DMRs, including *GRB10*:alt-TSS-DMR, but not at ICR1 or ICR2. *NAP1L5*:TSS-DMR and *WRB*:alt-TSS-DMR were also frequently hypomethylated in biparental-PMD specimens, but the genes associated with these DMRs may not be involved in the pathogenesis of biparental PMD because normal-sized placentas were found in mice with two paternal copies of *NAP1L5* and there was no correlation between *WRB* expression and DMR methylation in human placentas [79,80]. M, maternal allele; P, paternal allele. Pale blue shading indicates hypomethylation (<mean of normal − 2SD) and pink shading indicates hypermethylation (>mean of normal + 2SD). This figure was modified with permission from Aoki et al. [28].

The hypomethylation of multiple maternally methylated DMRs in biparental PMD evokes MLIDs in BWS and BiCHM. BiCHM is a rare familial form of CHM. Although CHM is androgenetic and generally sporadic, BiCHM shows a normal biparental diploid genotype, and the affected women suffer recurrent molar pregnancies. This condition displays maternal-effect autosomal recessive inheritance [81]. Approximately 75% of BiCHM cases are caused by pathogenic variants of *NLRP7*, and 5–10% of cases are caused by pathogenic variants of *KHDC3L* [67]. Recently, whole-exome sequencing (WES) has been performed on four biparental-PMD specimens, but no pathogenic variant was detected [28]. However, a homozygous missense variant of *PADI6* (c.1796T > A, p.Ile599Asn) was found in a patient who had had four hydatidiform moles (HMs) in a total of seven spontaneous pregnancies. Of these four HMs, two were diagnosed as PMD. One of these was diagnosed based only on the expression pattern of p57^KIP2^ in non-molar chorionic villi, and the other was diagnosed based on its microscopic morphology [82]. 

In BiCHM cases with pathogenic variants of *NLRP7* or *KHDC3L*, the maternally methylated DMRs were severely hypomethylated [83,84,85], suggesting that the origin of the aberrant hypomethylation is in the oocyte. However, the extent of the aberrant hypomethylation in biparental PMD was less severe than in BiCHM [28] and usually featured a normal fetus, indicating that the aberrant hypomethylation occurred postfertilization, resulting in mosaicism of aberrantly and normally methylated cells (Figure 2C).

All of this evidence points to the SCMC genes as the most important candidate for PMD pathogenesis. More PMD samples and mothers should be genetically analyzed to clarify the involvement of the SCMC genes and/or other genes in PMD pathogenesis, however.

#### 5.2.3. Other Possible Etiologies

Since more than 80% of fetuses born from PMD pregnancies thus far recorded have been female, the involvement of an X-linked gene has been suggested [22]. Vascular endothelial growth factor D (*VEGFD*) at Xp22.2 has been proposed as a candidate gene, since it has been found to be upregulated in the cystic areas of PMD specimens [76,86]. However, its involvement in PMD pathogenesis has not yet been proven.

In one case, paternal uniparental disomy of chromosome 6 (pUPD6), which is one of the causative alterations for transient neonatal diabetes mellitus, an imprinting disorder, was found in both the neonate and the PMD placenta [87]. Since the pUPD6 in the PMD specimen in this case was isolated, which meant that there was normal methylation of ICR2 and a normal biparental genotype except for chromosome 6, hypomethylation of *PLAGL1*:alt-TSS-DMR at 6q24.2 was a possible etiology for the PMD. However, no additional PMD cases with isolated pUPD6 have been reported since then.

### 5.3. Molecular Characteristics of BWS and PMD

To date, there have been only six cases in which both the BWS patient and the PMD tissues were molecularly analyzed (Table 1). The same molecular characteristics were found in three cases; two of these were ABM [28,88] and one was pUPD11 limited to 11p [32]. In the remaining three, the molecular characteristics of the BWS and the PMD differed. The first case showed no molecular alteration in the BWS neonate and ABM in the PMD specimen; the second showed ICR2-LOM in the BWS neonate and a biparental genotype in the PMD specimen; the last showed pUPD11 limited to 11p in the BWS neonate and a biparental genotype in the PMD specimen (Table 1). The results suggested that the cells of origin for the two conditions were the same in some cases but different in others. In the cases in which the molecular characteristics differed between the BWS and PMD, it is possible that those molecular characteristics arose at different times during the postfertilization period, and that these different molecular characteristics may be the critical factors determining cell fates, differentiation into extraembryonic tissue or embryonic tissue, or later retention or elimination in either tissue type [28].

## 6. Conclusions

Both PMD with ABM and biparental PMD, as well as BWS and CHM, are imprinting disorders. These disorders have some molecular characteristics in common, suggesting that PMD is the missing link between imprinting disorders occurring in liveborn children (BWS) and placental disorders that are incompatible with life (CHM and BiCHM) [28]. However, it remains unclear why PMD is common in BWS but not reported in other imprinting disorders. Three of the cases in Table 1 shared molecular alterations (ABM and pUPD11 limited to 11p) between BWS and PMD, suggesting the involvement of overexpression of *IGF2* and reduced expression of *CDKN1C* in both conditions. Aberrant imprinting of 11p15.5 may therefore be one of the reasons why PMD is common in BWS. However, the other three cases did not share molecular alterations between BWS and PMD, suggesting other as-yet-unknown causes. To clarify this, more precise and exhaustive genetic analyses, such as WES and whole-genome sequencing, should be performed not only on PMD specimens, but also on the mothers. In addition, whole-genome methylation analysis beyond the imprinted DMRs and other epigenome analyses should also be done. The results of these analyses may make it possible to diagnose PMD via noninvasive prenatal testing, which would be useful for clinical diagnostics and pregnancy and perinatal care.

## Figures and Tables

**Figure 1 cancers-14-05563-f001:**
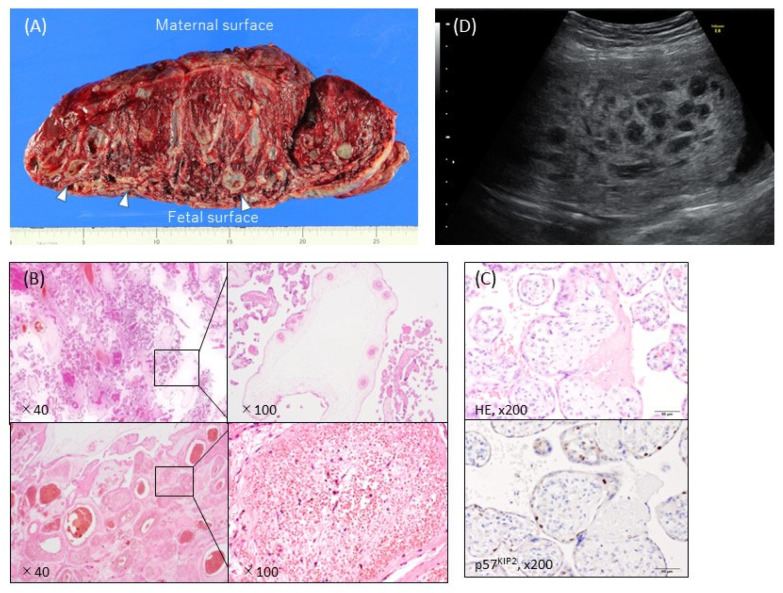
Typical ultrasonographic and macroscopic views and histopathological findings of placental mesenchymal dysplasia (PMD). (**A**) Macroscopic view of the incised plane. The placental weight was 1530 g at 38 weeks and 0 days of pregnancy. There were heterogeneous areas, with numerous cysts containing gelatinous liquid (arrows) and normal red-brown or spongy villous tissue. The cystic areas were predominantly on the fetal placental surface. (**B**) Hematoxylin-eosin stain. The PMD featured enlarged edematous stem villi with cisterns. Dilated, thick-walled, and thrombosed blood vessels with fibromuscular hyperplasia were also apparent. No trophoblastic proliferation was observed. (**C**) Immunohistochemistry against p57^KIP2^, which was not expressed in the stroma cells of dysplastic villi in the PMD placenta. (**D**) Ultrasonography revealed a thickened chorionic plate with a multicystic lesion and the living fetus (19 weeks and 2 days of pregnancy).

**Figure 2 cancers-14-05563-f002:**
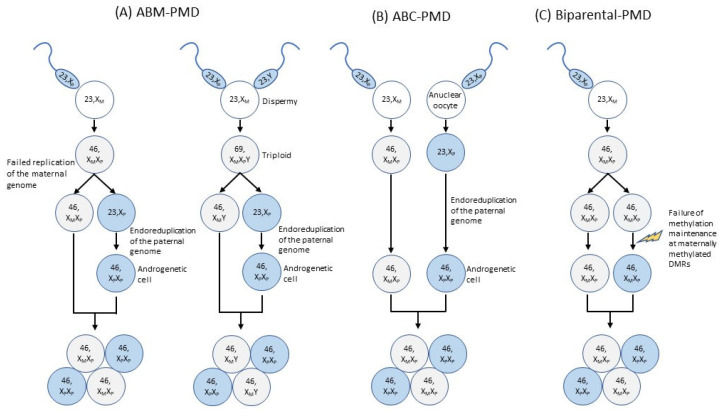
The etiologies of PMD. (**A**) Mechanism of PMD with androgenetic/biparental mosaicism (ABM). Left: Failure of maternal replication. ABM occurs when the zygote divides after the paternal genome has been replicated but the maternal genome has not. The daughter cell that carries only the paternal genome gives rise to androgenetic cells by way of endoreduplication of the paternal genome. Right: Dispermy. ABM occurs when a triploid zygote formed by two sperms and one oocyte divides without replication, giving rise to a (diploid) biparental cell and a haploid cell carrying only a paternal genome. The paternal genome is then endoreduplicated in the haploid cell, giving rise to androgenetic cells. (**B**) Mechanism of PMD with androgenetic/biparental chimera (ABC). ABC occurs when a normal zygote fuses with one created by the fertilization of an anuclear oocyte by a sperm cell. In the latter zygote, the paternal genome is endoreduplicated, giving rise to androgenetic cells. (**C**) Mechanism of biparental PMD. Normal fertilization is followed by normal cell divisions. Failure of methylation maintenance at maternally methylated differentially methylated regions (DMRs), including placenta-specific and ubiquitous DMRs, may occur during the postfertilization period, resulting in mosaicism of aberrantly and normally methylated cells.

**Table 1 cancers-14-05563-t001:** Molecular characteristics of BWS and PMD.

BWS			PMD					
Sex	Karyotype	Molecular Characteristics	Karyotype	PMD Genotype	DNA methylation of DMRs	Suspected Fertilization	Case no. or ID in the Publication	Reference
Female	46,XX	ABM (probable heterodisomy)	46,XX	ABM (probable heterodisomy)	NA	2 sperms	one case report	H’mida et al. [88]
Male	46,XY [17]/46,XX [2]	ABM (isodisomy)	NA	ABM (isodisomy)	NA	2 sperms	PMD-bws027	Aoki et al. [28]
Female	NA	Biparental with pUPD11 (limited to 11p)	NA	Biparental with pUPD11 (limited to 11p)	NA	1 sperm	4	Armes et al. [32]
Female	46,XX	No alteration	NA	ABM (isodisomy)	NA	1 sperm	PMD-022	Aoki et al. [28]
Male	NA	ICR2-LOM	NA	Biparental	Multiple aberrant methylation of both placenta-specific DMRs and ubiquitous DMRs	1 sperm	PMD-020 *	Aoki et al. [28]
Female	46,XX	pUPD11 (limited to 11p)	NA	Biparental	Multiple aberrant methylation of both placenta-specific DMRs and ubiquitous DMRs	1 sperm	PMD-bws022 *	Aoki et al. [28]

NA, not available. * These cases are illustrated in Figure 3.

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
