# Peer review of "Placental Mesenchymal Dysplasia and Beckwith–Wiedemann Syndrome"

_cancers, 2022, doi:10.3390/cancers14225563_

Round 1

Reviewer 1 Report (Previous Reviewer 1)

The authors have addressed all comments.

Reviewer 2 Report (Previous Reviewer 2)

The authors have appropriately responded to the reviewer's comments and modified their manuscript. I suggest accepting the revised manuscript.

This manuscript is a resubmission of an earlier submission. The following is a list of the peer review reports and author responses from that submission.

Round 1

Reviewer 1 Report

The review by Soejima and coauthors is comprehensive and well written. The structure is convincing and easy to follow. I´ve only minor remarks:

The nomenclature of the differentially methylated regions should follow the suggested standards, according to Monk et al. (PMID: 27911167).

I´ve difficulties with “variants in KCNQ1” as this might be misleading. The authors are right that variants in the genes are associated with aberrant IC2 methylation, but it is important to explain that the variants only cause these imprinting disturbances in case the KCNQ1OT1 is affected. KCNQ1 variants alone cause LongQT.

The term “genome-wide paternal UPD” is frequently used, but the authors should mention that the correct name should be paternal uniparental diploidy. It is interesting that the authors explain that ABM is the same.

A figure showing microscopic pathology and ultrasound findings would be helpful

Line 180: BWS is a frequent complication of PMD. Better: BWS is frequently associated with PMD

Figure 1: Probably I´ve missed it, but where did the authors get these data from, own cases?

Reviewer 2 Report

The authors reviewed Placental mesenchymal dysplasia (PMD) and Beckwith-Widemann syndrome (BWS) inks this paper is clinically important. It has been known that PMD is frequently observed in the placenta of cases with BWS, however, there are a few studies and reviews. Therefore, this review is important and useful. The reviewer has some comments and questions.

Comment 1

BWS often presents with placental hyperplasia; some BWS cases are complicated by PMD, while others present with only placental hyperplasia without PMD.

Is the histology of placental hyperplasia identified in BWS partially shared with PMD or is it a different mechanism?

Comment 2

The authors state in the Conclusion that "Both PMD with ABM and biparental PMD, as well as BWS and CHM, are imprinting disorders." The reviewer questions why PMD is so common in BWS and not reported in other imprinting anomalies. We would like some discussion on this point.

Comment 3

The etiology of PMD in 5.2 is easier to explain with a diagram.

Comment 4

I have looked at PubMed and it seems that there are reports of chromosomal aberrations not associated with imprinting abnormalities or with imprinting abnormalities outside the 11p15 region as an etiology for PMD. Since there are few reviews of PMD, it would be helpful for readers to understand PMD if other causes other than 11p15 are also described.

Comment 5

In the description of 5.2, the authors described as following; the following statement is made.”In PMD, androgenetic cells are distributed 296 throughout the chorionic membrane, chorionic mesenchyme, stroma, and enlarged chorionic vessels, but not the trophoblast, which contains only biparental cells [8, 29]. PMD trophoblasts, therefore, do not exhibit any abnormal proliferation, and they test positive for p57KIP2 (encoded by CDKN1C) [8, 29]. IGF2 is a growth-promoting factor and imprinted gene on 11p15, as well as CDKN1C. Have been previously examined for expression levels of IGF2 in trophoblasts in PMD, or are there any reports of PMD complications in BWS of ICR1-GOM?

Comment 6

In 5.3. Molecular characteristics of BWS and PMD, the authors described as follows; The same molecular characteristics were found in three cases; two of these were ABM. Does this mean that the molecular characteristics and PMD genotype in Table 1 are the same? The authors need more explanation.

Comment 7

The DMRs in DNA methylation of DMRs in Table 1 include both placenta-specific DMRs and ubiquitous DMRs. The authors need more explanation.
